# One Raft to Guide Them All, and in Axon Regeneration Inhibit Them

**DOI:** 10.3390/ijms22095009

**Published:** 2021-05-08

**Authors:** Marc Hernaiz-Llorens, Ramón Martínez-Mármol, Cristina Roselló-Busquets, Eduardo Soriano

**Affiliations:** 1Department of Cell Biology, Physiology and Immunology, Faculty of Biology and Institute of Neurosciences, University of Barcelona, 08028 Barcelona, Spain; cris.rb06@gmail.com (C.R.-B.); esoriano@ub.edu (E.S.); 2Clem Jones Centre for Ageing Dementia Research, Queensland Brain Institute, The University of Queensland, St Lucia Campus, Brisbane, QLD 4072, Australia; 3Centro de Investigación Biomédica en Red sobre Enfermedades Neurodegenerativas (CIBERNED), ISCIII, 28031 Madrid, Spain

**Keywords:** axonal regeneration, lipid rafts, cholesterol, sphingolipid, CNS injury, neurodegeneration, axonal growth-inhibitory molecules

## Abstract

Central nervous system damage caused by traumatic injuries, iatrogenicity due to surgical interventions, stroke and neurodegenerative diseases is one of the most prevalent reasons for physical disability worldwide. During development, axons must elongate from the neuronal cell body to contact their precise target cell and establish functional connections. However, the capacity of the adult nervous system to restore its functionality after injury is limited. Given the inefficacy of the nervous system to heal and regenerate after damage, new therapies are under investigation to enhance axonal regeneration. Axon guidance cues and receptors, as well as the molecular machinery activated after nervous system damage, are organized into lipid raft microdomains, a term typically used to describe nanoscale membrane domains enriched in cholesterol and glycosphingolipids that act as signaling platforms for certain transmembrane proteins. Here, we systematically review the most recent findings that link the stability of lipid rafts and their composition with the capacity of axons to regenerate and rebuild functional neural circuits after damage.

## 1. Introduction

The capacity of damaged axons to regenerate and re-innervate after injury is very limited in the adult central nervous system (CNS). In the 1980s, Aguayo and colleagues demonstrated that certain neuronal types are able to regrow their axons only in specific permissive environments [1]. Neuronal damage can be classified into two different groups: (i) traumatic lesions that are often caused by sudden mechanical forces, which results in the rapid displacement of the neural tissue and abrupt disruption of neural connections and (ii) nontraumatic lesions that are caused by neurodegenerative diseases such as Parkinson’s disease, Alzheimer’s disease or multiple sclerosis, also resulting in irreversible damage to nerve fibers. To regain functionality of the injured circuit, it is necessary to initiate the molecular mechanisms associated with two key processes: axon regeneration [2] and myelin sheath repair [3]. Unfortunately, the new extracellular environment that surrounds the damaged area has inhibitory effects on neuronal repair [4], where the presence of molecules such as myelin-associated inhibitors (MAIs) and chondroitin sulfate proteoglycans (CSPGs) curtails the ability of axons to regenerate. In addition to intrinsic features of the extracellular environment that prevent axonal regrowth, the release of several axon guidance molecules including netrins, semaphorins and ephrins also plays a pivotal role in axonal regeneration, either promoting or preventing axonal sprouting in lesioned nerves [5].

The peripheral nervous system (PNS) has a striking ability to regenerate axons. Experiments after PNS lesion revealed that CNS neurons are able to regenerate their axons in a PNS environment [1,6]. This led to the assumption that the limited capacity of regeneration of the CNS neurons is due to extrinsic cellular features in the damaged tissue. A key difference between the PNS and the CNS is the myelin composition, leading to the hypothesis that CNS myelin forms a non-permissive environment for axon regeneration.

In recent years, a multitude of signaling molecules and receptors have been shown to be associated with specific cell membrane microdomains termed lipid rafts. Lipid rafts are cholesterol- and glycosphingolipid-enriched nanoscale assemblies that associate and disassociate in a subsecond timescale (Figure 1) [7]. These microdomains act as signaling and trafficking platforms [8] and are important modulators of the activity of signaling pathways and effector molecules by either excluding or embedding these proteins into raft domains [9]. Since their discovery in the 1970s, a wide range of methodologies, from biochemical techniques to super-resolution microscopy, have allowed the obtention of a profound knowledge of these membrane microstructures (Table 1). In particular, the latest advances in the study of lipid rafts and their associated proteins pointed to the importance of these microdomains during axon regrowth after lesion. Manipulation of these membrane microstructures may become a potential therapeutic target to promote nerve repair. Lipid rafts differ in their lipid composition; however, there is a clear consensus about the enrichment in cholesterol and sphingolipids, and their biophysical properties. Lipid rafts commonly confer lateral heterogenicity to the cell membrane and are able to include or exclude certain membrane proteins whose functionality is regulated [10]. Post-translational modifications such as myristoylation, palmytoylation, or the incorporation of glycosylphosphatidylinositol (GPI) groups are common between raft-resident proteins [11].

In this review, we describe the extrinsic molecular machinery involved in axon regeneration whose activity is dependent on lipid raft association, as well as experimental procedures to manipulate and comprehend the role of membrane raft microdomains in the regeneration of injured nerves.

## 2. Extracellular Matrix Molecules

### 2.1. Chondroitin Sulfate Proteoglycans (CSPGs)

CSPGs are extracellular matrix (ECM) molecules widely expressed in the CNS during development and are a family of glycoproteins that share a common structure formed by a protein core and side chains of chondroitin sulfate, which are sulfated glycosaminoglycan (GAG) sugars. These molecules act as guidance cues during cell migration and axon elongation during brain development, as well as modulators of synaptic connections in adult brains [24]. However, CSPGs act as inhibitory ECM molecules that are highly expressed after CNS damage, being one of the most important known components that severely restrict axonal elongation after injury. After CNS damage, a structure enriched in reactive glial cells known as the glial scar is formed, surrounding the damaged area and reducing the spreading of the inflammatory response to other regions by delimiting the site of injury [25]. Elevated levels of CSPGs expressed by activated astrocytes are also found in neurodegenerative diseases [26]. 

Although the ability of CSPGs to restrict axonal regeneration and plasticity has been well documented [27], the molecular mechanisms by which CSPGs hinder neurite outgrowth are poorly understood. Studies on the degradation of GAG chains by chondroitinase ABC treatment have shown that the inhibitory nature of CSPGs partially relies on the integrity of GAG [28], as digestion of the GAG side chains confers the glial scar a more permissive environment after injury, which promotes axon elongation [29]. To date, a number of CSPGs receptors have been identified in the cell membrane of neurons, and they are the main cause by which axon regeneration is restricted after CNS injury. These molecules establish a non-permissive environment by interfering with the growth-promotion action of adhesion molecules and by facilitating the function of some chemorepulsive cues. The transmembrane protein tyrosine phosphatase PTPσ binds with high affinity to CSPGs, causing axon retraction after CNS damage [30]. The Nogo Receptor 1 and 3 (NgR1 and NgR3, respectively) are GPI-linked membrane protein receptors for three myelin-associated inhibitors (MAIs), Nogo, Myelin Associated Glycoprotein (MAG) and oligodendrocyte-myelin glycoprotein (OMgp) [31,32,33]. NgR1 and NgR3 have also been shown to interact with CSPGs to promote the inhibition of axon regeneration [34].

Lecticans are members of the CSPG family. One of the major lecticans in the CNS is Aggrecan, which is abundant in the developing and injured CNS. Lectican is known to inhibit axon regeneration and neurite extension through structures located on the core glycoprotein, rather than the CS chains [35]. Interestingly, Brevican, another member of the lectican family not exclusively found in the ECM, is also linked to the cell membrane via a GPI-anchor [36], being associated with raft domains. 

### 2.2. Heparan Sulfate Proteoglycans (HSPGs)

HSPGs are another type of ECM glycoprotein that, similarly to CSPGs, are expressed in the CNS during developmental stages [37] and in the adult CNS only after injury [38]. However, and contrarily to CSPGs, HSPGs play a role as growth-supportive molecules. Glypican-2 is a GPI-anchored HSPG highly expressed in the mammalian CNS during neuritogenesis [39] that acts as a receptor of the differentiation factor heparin-binding growth-associated molecule (HB-GAM). In regeneration studies, HB-GAM injection after spinal cord injury (SCI) has been shown to reverse the inhibitory effects of CSPGs through its binding to Glypican-2 in cholesterol-enriched microdomains [40]. On the contrary, the homologous Glypican-1 serves as a high-affinity receptor for Slit proteins [41], and it is highly expressed in reactive astrocytes after injury [42]. Slit2 and Glypican-1 are highly expressed and have been shown to interact in experimental models of brain injury, preventing regenerating axons from entering the lesioned area [42].

Syndecan is another HSPG that is necessary for growth cone stabilization after axotomy in *C. elegans*. In this model, Syndecan plays a canonical function in axon guidance that relies on the expression of HS chains and the Syndecan core protein, which exert an intrinsic function in growth cone stabilization [43]. Moreover, Syndecan knockdown in cultured DRG neurons induces a reduction in neurite extension [44]. Interestingly, Syndecan has been shown to be preferentially distributed into lipid rafts microdomains after growth factor stimulation [45].

Both CSPGs and HSPGs bind to the receptor-type protein tyrosine phosphatase PTPRσ, which affects axon regeneration [46]. The patterns of sulfate and sulfamate groups within the CS and HS determine the binding to PTPRσ [46]. CSPGs activate PTPRσ and disrupt the autophagy flux at the autophagosome, sufficient for dystrophic end ball formation and the consequent inhibition of axonal regeneration, which can be reversed with the addition of HS oligosaccharides holding sulfate and sulfamate groups, resulting in receptor clustering [46].

### 2.3. Tenascins (TNs)

TN glycoproteins are expressed by oligodendrocytes, interneurons and motoneurons and are characteristic constituents of the perineuronal net (PNN), the reticular scaffold of ECM molecules that surrounds neuronal cell bodies, dendrites and the axonal initial segment of some neuronal populations in the CNS, including the visual cortex, the deep cerebellar nuclei, the substantia nigra, the hippocampus and the spinal cord [47]. Two of the known TN proteins are TN-C and TN-R, which are expressed in the brain and are developmentally regulated [48]. Together with CSPGs, TNs are highly expressed in the glial scar after CNS damage [49]. After spinal cord injury in rats, TN-C expression is detected in the scar just 24 h post-injury, with peak levels observed 7 days later [49]. In the mice developing olfactory bulb, TN-C is expressed in a boundary-like pattern, inhibiting olfactory sensory neuron axons to grow beyond this region [50]. Thus, axonal extension remains restricted to the olfactory nerve layer, being crucial for axon sorting during development [51]. TN-R mediates its inhibitory neurite outgrowth through the cell surface receptor F3/contactin [52]. F3/contactin is a well-characterized GPI-anchored molecule abundantly found in lipid raft microdomains [53]. Antibody cross-linking of F3/contactin in lipid rafts activates Fyn in the inner membrane leaflet [54], which has been associated with growth cone collapse [55]. TN-C is detected in brain raft microdomains [56], and both TN-C and TN-R are distributed in membranes of mice cerebellar oligodendrocytes, forming patches that are reduced after the incubation of the lipid raft destabilizing agent, MβCD [56]. The lipid raft association of TN and its receptor F3/contactin manifests the importance of their specific membrane localization in the signaling associated with the arrest of the growth cone extension. Taking into account that TN glycoproteins are highly expressed by oligodendrocytes in the vicinity of the damaged CNS, the association of these proteins to lipid rafts are of great potential in tackling the restricted axon growth after nervous tissue injury.

Integrins consist of α and β chains and are heterodimeric receptors of ECM molecules and other ligands. Several studies have addressed the importance of lipid rafts in the regulation of integrins signaling. For instance, the activation and clustering of α4β1 integrin is raft-dependent [57]. β1 integrin has been found to be distributed in membrane rafts. Furthermore, MβCD inhibits the cell adhesion dependent on integrin-fibronectin interaction [58]. Thus, some integrin subunits may rely on the association to lipid rafts in cell adhesion and signal transduction processes, which are key features during axon regeneration in non-permissive environments. Interestingly, the expression of TN-C binding integrin subunits enhances the capacity of axons to regenerate in a TN-C rich environment, such as the injured spinal cord [59]. α9β1 integrin is able to specifically bind to TN-C [60]. Both in vitro and in vivo studies have shown that α9 integrin, an absent subunit in adult neurons, promotes axonal regeneration [59]. By overexpressing the α9 integrin subunit, TN-C inhibitory signaling can not only be prevented but can also be used as a gene therapy strategy to enhance axonal regeneration in SCI models.

## 3. Myelin-Associated Inhibitors (MAIs)

Research efforts in recent years have focused on defining the molecular players within the myelin sheaths curtailing potential axon regeneration, establishing the concept of MAIs. To date, three prototypical MAIs have been identified: myelin-associated glycoprotein (MAG), oligodendrocyte myelin glycoprotein (OMgp) and Nogo. Each of the three identified MAIs signal through different receptors located in the cell membrane of neurons.

### 3.1. Myelin-Associated Glycoprotein (MAG)

MAG is a transmembrane glycoprotein produced by oligodendrocytes in the CNS [61]. Some discrepancies exist in the function of MAG in response to injury and disease. Genetic deletion of MAG reduced corticospinal tract (CST) axon sprouting, an important tract widely studied in SCI models [62]. Conversely, MAG has been extensively used as an inhibitory substrate for axon elongation in mature neurons [63].

Gangliosides are glycosphingolipids that carry one or more sialic acid residue(s). A particularity of brain gangliosides is that due to the biophysical properties of their ceramide moiety, they tend to aggregate in lipid microdomains [64]. Among the gangliosides expressed in the brain, GD1a and GT1b have been shown to interact with MAG to induce the inhibition of axon regeneration [65]. Both gangliosides reside in membrane rafts, where they interact with the GPI-linked NgR2 protein to inhibit axon outgrowth after CNS damage [66]. Isolated MAG from whole-brain extracts and oligodendrocytes are found in lipid raft compartments [67]. Localization within these domains is dependent on cell membrane cholesterol, as demonstrated after MβCD treatment [67]. Moreover, the localization of NgR1 and GT1b within cholesterol-enriched membrane microstructures is required for the interaction of MAG and these receptors [67]. This molecular environment may facilitate the downstream signaling required for the response to MAG. Furthermore, MAG or the antibody against GT1b and GD1a promote the recruitment of p75NTR to lipid rafts, suggesting the importance of these gangliosides in the recruitment of p75NTR to cholesterol-enriched microdomains [68]. Disruption of lipid rafts abolishes the inhibitory effects induced by MAG and Nogo peptides in postnatal cerebellar neurons, indicating that raft microdomains play an important role in the initiation of its signal transduction during neurite outgrowth [68]. MAG also binds to the lipoprotein-receptor-related protein-1 (LRP1), which is involved in the activation of RhoA, thereby facilitating the collapse of growth cones and inhibiting neurite outgrowth [69]. MAG-mediated activation of RhoA may involve both LRP1 and p75NTR, as its binding induces colocalization of both receptors [69]. In neurons and neuronal-derived cell lines like PC12 and N2a cells, LRP1 has been found to partially localize in lipid rafts [70]. Upon raft disruption with MβCD, LRP1-mediated signaling is impaired, while its ligand-binding capacity remains intact [70]. This indicates that MAG plays an important role in recruiting LRP1 and p75NTR receptors to lipid rafts, where the MAG-dependent signaling cascade that promotes axon retraction is initiated. Therefore, targeting lipid raft stability can become a novel strategy to prevent axonal retraction after damage induced by MAG and their receptors.

Many pieces of evidences suggest that integrins participate in the regulation of neurite extension, axonal guidance and neuronal migration [71]. β1 integrin has been shown to act as a receptor for MAG, mediating growth cone responses independent of NgRs in rat hippocampal neurons. β1 integrin signaling has been demonstrated to be required for both MAG-induced attraction in developmental stages of the CNS and repulsion in mature CNS [72].

### 3.2. Oligodendrocyte Myelin Glycoprotein (OMgp)

OMgp is a GPI-anchored protein and a ligand of the NgR protein. OMgp is expressed not only by oligodendrocytes but also by neurons in the adult CNS [73]. This protein was shown to inhibit axon regeneration after CNS injury in vivo because its deletion enhances axon sprouting in SCI models [74]. OMgp is anchored solely in the outer leaflet of the plasma membrane through its GPI linker. Detergent solubility assays have demonstrated that OMgp is a lipid raft component in myelin mouse extracts and that associates with caveolin-1 [75].

### 3.3. Nogo

Nogo is a transmembrane protein expressed by oligodendrocytes, which is known to induce growth cone collapse and neurite retraction through its extracellular 66 amino acid loop [76]. As previously reviewed in this article, Nogo has been found to be closely associated with lipid raft microdomains, as well as their receptors and coreceptors, NgR and p75NTR, a prerequisite to inhibit neurite outgrowth [77,78].

NgR and p75NTR are the receptor and coreceptor of Nogo, respectively, and have been shown to reside in lipid raft microdomains [77]. Their inhibitory function can be hampered after destabilizing lipid rafts in cells treated with methyl-β-cyclodextrin (MβCD), a pharmacological reagent that forms inclusion complexes with membrane cholesterol [79]. In the cell membrane of the cerebellar granule cells, while the coreceptor p75NTR can be found in raft and non-raft regions, NgR receptors are located within cholesterol-enriched microdomains [77], where the interaction between these two proteins takes place [78]. The MAI Nogo-A acts as a potent neurite growth inhibitor that represses axonal regeneration and structural plasticity in the adult CNS [80]. This Nogo-induced neurite growth inhibition involves the Rho-associated protein kinase (ROCK) [81], a kinase that also localizes with raft membrane microdomains [82]. After MβCD treatment, Nogo stimulation fails to promote neurite inhibition in cerebellar granule cells due to abrogated activation of RhoA [77]. Recent studies using specific function-blocking antibodies against Nogo have been shown to improve locomotion recovery in rodents after spinal cord injury [83]. Thus, dodging neurite outgrowth inhibition by halting the signaling cascade triggered by Nogo and their receptors NgR and p75NTR represents a promising strategy to enhance axon regeneration after CNS damage.

Nogo-A also interacts with integrins, which are the most important cell surface transmembrane receptors of ECM proteins [84]. Integrins are involved mainly in developmental processes such as migration, adhesion or axon growth and are highly downregulated in the mature CNS [85]. Integrins have been studied as pivotal players during axon regeneration in the CNS [71,85,86]. PNS neuronal types that express relatively high levels of integrins [87], such as dorsal root ganglion (DRG) neurons, have been shown to regenerate most readily, a feature that might be associated with the enhanced ability of the PNS to regenerate after injury [88]. Through the interaction of Nogo-A with integrins, Nogo-A suppresses the integrin signaling by inactivating them in vitro [86] and in vivo [89]. Nogo-A interferes with the function of fibronectin-associated integrins, resulting in the attenuation of the neurite outgrowth in DRG neurons [84]. Moreover, after optic nerve crush, Nogo-A downregulates the expression of αV integrins, which results in the inhibition of the phosphorylation of the Focal Adhesion Kinase (FAK) [89], and consequently halts the axonal regeneration.

## 4. Axon Guidance Cues

Over the decades, several studies have focused on the role of guidance molecules and the signaling pathways involved in axonal growth and pathfinding during development. However, little is known about their role in the adult CNS when it comes to repair and regrowth axons to achieve functionality. A number of axon guidance cues follow a determined pattern of expression after CNS injury, to which injured axons are responsive. Interestingly, some of these molecules and their receptors exert inhibitory effects in the regenerating axons. 

### 4.1. Netrins

Netrin-1 was the first axon guidance cue to be discovered in vertebrates. Netrin-1 is a bifunctional cue that acts as a chemoattractive molecule through the Deleted in Colorectal Cancer (DCC) receptor and the Neogenin receptor, or as a chemorepellent through the Uncoordinated receptor 5 A-D (UNC5A-D) (Figure 2A) [90]. More recently, Down Syndrome Cell Adhesion Molecule (DSCAM) has also been shown to be a Netrin-1 receptor [91]. Both DCC and UNC5 receptors are dependence receptors, meaning that their expression creates a cellular state depending on the availability of their ligand. DCC is able to trigger apoptotic signaling upon cleaving of its intracellular domain when it is not interacting with Netrin-1 [92]. UNC5 receptors display a death domain cassette in their intracellular domain, equivalent to the one observed in other dependence receptors such as p75NTR. Similarly to DCC, UNC5 receptors are able to trigger apoptosis signaling, which is blocked by the presence of Netrin-1 [93]. Altogether, these receptors not only play a key role in axon guidance processes but also act as survival factors.

Many pieces of evidence have shown the importance of Netrin-1 in axon regeneration processes after CNS and PNS injury [94]. After SCI, overall Netrin-1 expression is dramatically reduced, but it is still expressed by neurons and oligodendrocytes immediately adjacent to the lesion area [95]. UNC5 and DCC levels are also reduced in the damaged spinal cord. However, UNC5 levels recover with time, while DCC levels remain at 50% of pre-injury values [95]. Netrin-1 expression patterns after SCI are similar to those of other MAIs in CNS lesion models [96]. The dominant expression of the UNC5A-C receptors and the persistent expression of Netrin-1 support the hypothesis that regenerating axons are not able to enter the damaged tissue due to the hostile environment produced by Netrin-1 and its receptors UNC5.

Interestingly, several studies have demonstrated that Netrin-1 receptors reside in lipid raft microdomains. The specific localization of UNC5 and DCC in these regions appears to be necessary for cell death signaling and axon guidance events. The DCC/Netrin-1 pair regulates neuron survival during nervous system development, conferring a dependent state to those cells expressing this receptor [97]. Localization of DCC to lipid rafts microdomains is requisite for its apoptotic activity in immortalized cells and primary neurons [97]. Axon attraction elicited by Netrin-1 through its receptor DCC has also been found to depend on lipid raft localization. The embedding of DCC in cholesterol-enriched microdomains depends on the palmitoylation in its transmembrane region [98]. Destabilization of membrane microdomains with MβCD or Cholesterol Oxidase (ChOx), or mutation of the amino acid residue that is palmitoylated, prevents the association of DCC to lipid rafts and the Netrin-1-mediated axon outgrowth [98]. UNC5 receptors are also embedded in cholesterol-enriched microdomains, and this specific membrane localization is necessary for their proapoptotic functions when unbound to Netrin-1 [99]. Presumably, lipid raft targeting is mediated through adaptor proteins within their intracellular death domain [99]. The interaction between the UNC5B receptor with the pro-apoptotic death-associated protein kinase 1 (DAPK1) depends on the localization into lipid rafts [99]. Similar to the DCC receptor, association with raft membranes has also been proven to be essential during axonal guidance events, as shown not only by biochemical fractionation, but also with biophysical tools. Using Fluorescence Recovery After Photobleaching (FRAP) and single-particle tracking Photoactivated Localization Microscopy (PALM), we have demonstrated that UNC5 receptors reside in raft membrane microdomains [100]. Such properties are modified after MβCD or cholesterol oxidase (ChOx) treatment [100]. Destabilization of lipid rafts in primary neuronal cultures and brain cerebellar explants inhibits the chemorepulsive response of growth cones and axons against Netrin-1 [100].

### 4.2. Ephrins

Ephrins are membrane-bound molecules that work as the ligands of the Eph (erythropoietin-producing hepatocellular carcinoma) tyrosine kinase receptors, which represent the largest family of receptor tyrosine kinases. To date, 14 Eph receptors have been identified in vertebrates, divided into two subgroups, EphA and EphB. All Eph receptors are composed of seven conserved structural domains. An N-terminal domain is located in the extracellular part including the Ephrin-binding domain, forming a globular β-barrel [101]. There are nine different types of ephrins, which can be subdivided into two groups: ephrinA and ephrinB. All ephrins share a conserved extracellular domain but differ in their attachment to the cell membrane. EphrinAs are linked through a GPI group at their C-terminus, whereas ephrinBs contain a single transmembrane and a short cytoplasmic domain. Eph receptors and ephrin ligands have been found to be implicated in numerous processes during CNS development, such as delimiting migration patterns of neural crest cells, boundary formation between different hindbrain segments and the proper formation of CST [102].

Both Eph receptors and ephrin ligands are membrane-attached molecules that, when interacting in *trans*, initiate a bidirectional signal transduction that will propagate simultaneously in both contacting cells (Figure 2B). Thus, signal transduction occurs bidirectionally, influencing the behavior of both interacting cells [103]. The predominant neural response to ephrins is repulsive [5]. Activation of the EphA receptors by clustered ephrinAs initiates a repulsive response leading to growth cone turning or collapse. Small GTPases of the Rho family link the EphA receptors to actin/microtubule dynamics. RhoA activation and its downstream effectors ROCK induce growth cone collapse and retraction [104]. After SCI in rats, EphA4 receptors tend to accumulate in sectioned CST axons, suggesting a responsiveness to ephrinB ligands expressed in the myelin, which results in the inhibition of regeneration [105]. Moreover, upregulation of EphA3 and EphA7 receptors in astrocytes after SCI [106] is suspected to be responsible for inducing the apoptosis of these cells [107], which results in a second prolonged wave of cell death, responsible for the secondary damage following trauma in the CNS.

Class A ephrins are bound to the cell membrane through a GPI anchor, facilitating their embedding into plasma membrane lipid rafts [108]. EphrinA2 and A5 have been shown to signal through lipid rafts when bound to their Eph receptors [109], and ephrinA-dependent retraction of retinal ganglion cell axons involves cAMP signaling restricted to lipid raft microdomains [110]. Transmembrane ephrinB ligands have also been shown to be localized in lipid raft discrete microdomains [111]. Stimulation of ephrinB1 with EphB receptors induces the reorganization of membrane rafts-containing ephrinB1 into larger raft microdomains. These larger microdomains concentrate glutamate receptor-interacting proteins (GRIP) in the cell membrane, resulting in the formation of a scaffold for the assembly of a multiprotein signaling complex downstream of ephrinB1 ligands [111]. 

### 4.3. Semaphorins

Semaphorins comprise a large family of more than 20 different molecules. They are widely expressed in the developing CNS, being crucial not only as axonal guidance cues but as key regulators of cell migration, cell death and synapse formation [112]. Semaphorins can be soluble proteins, transmembrane proteins or GPI-anchored proteins that localize in rafts. Semaphorins mainly signal through the Neuropilin (NRP) receptors and the Plexin receptors (Figure 2C). Stimulation of NRP1-expressing endothelial cells with Sema3C results in the recruitment of NRP1 receptors in lipid rafts, and their subsequent raft-dependent internalization. Therefore, it is likely that their internalization via rafts facilitates the coupling of NRP1 and its associated signaling partners to downstream effectors, resulting in the contribution to a different signaling specificity depending on their membrane association [113]. After inducing ischemia in the rodent brain, NRP1 translocates from non-raft domains to lipid rafts, together with Fer kinase, which regulates axon extension inhibition, and collapsing response mediator proteins (CRMPs) [114], known to mediate NRP1 response to Sema3A during axonal retraction of DRG neurons [115].

Sema3A, the most studied Semaphorin to date, is a soluble repulsive guidance molecule. It has been demonstrated that growth cone repulsion by Sema3A can be blocked after MβCD incubation in *Xenopus laevis* spinal cord neurons. Turning assays demonstrated that upon lipid raft destabilization, Sema3A stimulation did not result in a clear chemorepulsive response [116]. CRMPs have been shown to be intracellular messengers required for the growth cone collapse induced by Sema3A [117]. Interestingly, CRMPs are distributed in a punctate fashion in neurites and growth cones in cortical neurons, and this punctate distribution is altered after MβCD treatment [118]. CRMPs can be isolated by membrane fractionation in detergent-resistant fractions after solubilization of brain membranes [118]. Thus, the molecular machinery involved in the collapsing of growth cones induced by Sema3A is tightly dependent on the integrity of lipid rafts [118]. Moreover, in leukemic T cells, Sema3A has been demonstrated to trigger a proapoptotic program after its receptor plexinA1 is stimulated. Activation of plexinA1 results in the recruitment of Fas into lipid rafts microdomains, where the apoptotic cascade is originated. Disruption of lipid rafts reduces sensitivity to Fas-mediated apoptosis [119]. Cell death signaling is a hallmark of CNS damage [120] that needs to be overcome in order to preserve the integrity of the affected neural population. In these studies, Sema3A has been found to trigger cell death from lipid rafts microdomains in blood cell lines. Sema3A is highly expressed in the PNN, which has been shown to interact with CSPGs [121]. This association might be responsible for regulating the repulsive function of Sema3A in neurodegenerative processes [47,122]. Further investigations to test whether Sema3A induces apoptosis from lipid rafts in neurons may be a promising approach to avoid cell death after CSPGs expression in the damaged area of the nervous tissue. Demonstration of Sema3A-induced apoptosis from lipid rafts would represent a promising approach to overcome cell death derived from CNS damage.

Semaphorins are highly expressed in the glial scar after injury, together with the plexin and NTR receptors [123]. Many investigations have targeted the association between Sema3A and its receptors NRPs and plexin to enhance axon regeneration by blocking their inhibitory effects [124,125]. Association of Semaphorins and their receptors to lipid raft microdomains requires further investigations in order to design new regeneration therapies to control their function by manipulating the stability of these microdomains.

### 4.4. Slits

Slits are secreted proteins that bind to Roundabout (Robo) receptors. Robo receptors are expressed in the growth cones of developing commissural axons in the spinal cord and are well known for mediating axon repulsion in the developing CNS (Figure 2D). Expression of Slits in the floor plate of the developing spinal cord prevents Robo-expressing axons from re-crossing the midline [126]. Furthermore, Robo1 is able to bind the Netrin-1 DCC receptor, resulting in the inhibition of Netrin-1 attraction (Figure 2D) [127]. As stated previously, axonal attraction exerted by Netrin-1 through the DCC receptor relies on the association of the DCC to lipid raft microdomains [98]. Thus, further investigation is needed to decipher the importance of lipid raft integrity on harboring the interaction of Robo1 and DCC receptors. 

The involvement of Slit-Robo signaling during CNS regeneration remains unclear. Slits and their Robo receptors are upregulated after CNS damage, apparently being involved in the regenerative failure of CNS axons by either inhibiting axon outgrowth or by participating in the formation of the glial scar [128]. Neural stem cells (NSCs) transplantation is a promising therapy to promote neural regeneration in a damaged CNS [129]. In a mouse model for ischemic stroke, neuroblast migration is restricted by reactive astrocytes located in the lesioned area. These neuroblasts use Slit1-Robo2 signaling to migrate near the injury [130]. Interestingly, transplanted Slit1-overexpressing neuroblasts migrate closer to the poststroke lesion than non-overexpressing neuroblasts, regenerating neuronal circuits and resulting in partial functional recovery [130]. In vitro, silencing of Slit2 in NSCs inhibits cell proliferation and migration and enhances axon outgrowth [131]. 

Several studies suggest that Slits can bind to Glypican-1, a GPI-anchored heparan sulfate that resides in lipid rafts microdomains (Figure 2D) [132]. All Slit family members are found to be expressed in reactive astrocytes together with Glypican-1 after cryo-injured brain, a procedure consisting of stereotactical injection of liquid nitrogen in the brain that is known to mimic traumatic brain injury in humans, thereby suggesting that both types of proteins may have a role in preventing regenerating axons from entering into the lesioned area [42]. Moreover, in a mouse model of SCI, high mRNA levels of Glypican-1 and Robo3 receptor are detected at the site of injury. This upregulation of their expression is believed to increase the sensitivity of regenerating axons towards repulsive axon guidance cues [133]. Whether the interaction between Glypican-1 and Slit occurs specifically in lipid rafts is still unknown. However, it is plausible to think that these microdomains may play an important role in the signaling of Slit molecules after CNS damage, as Glypican-1 is a GPI-anchored molecule that is highly upregulated, together with Slits and Robo receptors.

## 5. Modulation of Membrane Lipids to Enhance Axonal Regeneration

Lipid rafts are cell membrane microstructures enriched in cholesterol and sphingolipids, which are important for many cellular processes (Figure 1). The integrity of lipid rafts is crucial for a multitude of signaling pathways in the CNS, such as cell migration, axon guidance and neuronal plasticity. Aberrant composition of membrane lipid rafts can lead to abnormal neurological functions and debut of neurodegenerative diseases [134]. Many of the elements that curtail the regeneration of axons after the injury of the CNS (Nogo, MAIs, CSPGs and guidance cues) are tightly linked to lipid rafts microdomains and rely on their specific membrane localization to be functional. Thus, systemically targeting and depleting lipid rafts may open the possibility of novel therapies to enhance axon regeneration, circuitry establishment and functional recovery of transplanted cells in injured brains. Different lipid raft destabilizing agents, summarized in Table 2, have been used over the years to study these membrane microstructures. Targeting either cholesterol or sphingolipid moieties opens a wide range of possible mechanisms to disrupt raft microdomains.

Because the main components of lipid rafts are cholesterol and sphingolipids, the majority of the efforts to overcome the inhibitory signaling after CNS damage have been focused on the disruption of these components, as summarized in Table 3. Neogenin, for example, promotes neural apoptosis in the absence of its ligand, the repulsive guidance molecule A (RGMa). However, when bound to RGMa, Neogenin initiates the repulsion of the axons in a process dependent on its association with lipid raft microdomains [148]. The reduction of membrane cholesterol hence blocks Neogenin raft localization, promotes axonal outgrowth and prevents neuronal apoptosis in injured adult optic nerve and the spinal cord [148]. Moreover, providing RGMa to neurons after stroke blocks Neogenin-induced neuronal death [149]. A newly designed human anti-RGMa antibody prevents the localization of Neogenin to lipid rafts, protecting the CNS from further ischemic damage, together with an increment in the neuronal network complexity, resulting in a significant improvement in functional recovery [149]. Cholesterol synthesis inhibition with Lovastatin or with the siRNA against the enzyme 7-dehydrocholesterol reductase inhibits the association of Neogenin to lipid rafts and enhances axonal growth in inhibitory myelin and RGMa substrates [150]. Furthermore, both strategies to dampen cholesterol synthesis allow for robust axonal regeneration and neuronal survival after optic nerve crush [150]. Prominin-1 has been identified as a regulator of the axon regeneration program. Elevated Prominin-1 levels in cultured DRG neurons enhance axon regeneration [151] and induce a dramatic down-regulation of cholesterol synthesis associated genes, with a consequent reduction in cholesterol levels [151].

Statins are widely used cholesterol-lowering drugs. Several types of statins were tested for their ability to enhance axon regeneration, and cerivastatin was found to be the most potent by increasing neurite outgrowth in a MAG or CSPGs enriched inhibitory substrates [152]. Inhibitory effects were retrieved upon addition of mevalonate, a key component of cholesterol biosynthesis inhibited by statins [152]. Cholesterol depletion and cholesterol biosynthesis inhibition cause an increment in the size of growth cones, the density of filopodium-like structures and the number of neurite branching points of developing neurons isolated from the CNS and the PNS [153]. Cholesterol depletion with Nystatin results in enhanced capacity of axon regeneration in axotomized neurons in vitro [153]. Furthermore, axonal regeneration and functional recovery can be achieved in a model of sciatic nerve axotomy after cholesterol depletion by MβCD [153]. Similarly, acute treatment of Nystatin enhances growth cone area in cultured hippocampal neurons, while chronic treatment enhances axon length, axon branching and axon regeneration post-axotomy [153]. Interestingly, depending on the duration of the treatment with Nystatin, distinct signaling pathways are activated. After acute treatment, Nystatin promotes growth cone expansion through Akt phosphorylation; meanwhile, chronic treatment results in an increment of nitric oxide levels, responsible for an additional enhancement of axon regeneration [153].

Not only myelin-associated proteins but also myelin lipids are involved in the creation of an inhibitory environment for axon regeneration. Cholesterol and sphingomyelin, two essential lipid components of myelin, operate through a Rho-dependent mechanism, having inhibitory effects on neurite outgrowth in different neuronal types. Thus, decreasing the levels of membrane cholesterol and sphingomyelin using 2-hydroxypropyl-β-cyclodextrin leads to increments of axon regeneration after SCI [154].

## 6. Conclusions

CNS regeneration has emerged as a pivotal landmark in the treatment of neurodegenerative diseases and injuries, such as stroke or trauma. The restricted capacity of the CNS to regenerate is one of the main challenges to overcome in order to achieve full functional recovery. Axons of damaged neurons fail to regrow mainly because they do not retain developmental characteristics, and because glial scar formation produces highly inhibitory molecules that curtail the axonal extension.

In recent years, many strategies have been developed to enhance CNS regeneration. Therapeutic approaches are mainly focused on blocking inhibitory components present in the glial scar, neuronal replacement and axon growth enhancement. These strategies comprise the design of inhibitory compounds to block the Nogo receptors or CSPGs [83,155], the application of new cell-based therapies to replace the damaged tissue, the promotion of neuroprotection, modulation of the immune response [156,157,158], and the use of bioactive materials that will be combined with cells and soluble molecules to recreate developmental environments that enhance neuronal regrowth and restore its functionality [159,160,161]. Among the limitations of neural transplantation therapies are the poor rate of survival, as well as the lack of plasticity and integration of transplanted cells. In this review, we have highlighted that lipid rafts not only are important for the inhibitory signals of axon regeneration but also for the apoptosis signaling cascades that take place after CNS injury. Treating these cells with lipid raft-modifying agents before transplantation could be a promising approach to facilitate their implantation, improve their functional rewiring, and increment the rate of survival of implanted cells. It has been demonstrated that lipid rafts are crucial for many cell signaling processes.

There is little knowledge about the differential distribution and composition of lipid rafts in different types of neurons, axonal and dendritic compartments and the changes during maturation of the CNS. Neurons are highly polarized cells, and signaling molecules that play a role in the specification of the somatodendritic and axonal compartments also reside in lipid rafts [162]. Lipid rafts are highly abundant in dendrites of cultured hippocampal neurons, and a variety of postsynaptic proteins are associated with them [163]. Disruption of lipid rafts leads to depletion of excitatory and inhibitory synapses, as well as the loss of dendritic spines [163]. Lipid rafts are also abundant in the growth cone, being important for axon guidance and elongation, as a great variety of molecules implicated in these processes are associated with lipid rafts [164,165]. Moreover, recent investigations have shown that deregulation of lipid homeostasis and lipid raft instability may negatively impact the aging brain and consequently the development of neurodegenerative diseases [134,166].

It has been demonstrated that lipid raft destabilization contributes to axon regeneration. However, it must be taken into consideration that other raft-associated molecules may also be affected by alterations in lipidic composition. Thus, further studies are needed in this direction to establish the suitable conditions of lipid raft modifications without compromising their cellular functions but enhancing neural regeneration. Nevertheless, this could be a valuable approach to overcome the limitations of cell-based therapies used to treat CNS injuries and neurodegenerative diseases.

In recent years, interest in intrinsic molecular mechanisms that drive axon regeneration has substantially grown [167]. Some of these intrinsic mechanisms are tightly related to intracellular lipid metabolism. For instance, deletion of the phosphatase and tensin homolog (PTEN), a negative regulator of the mammalian target of rapamycin (mTOR), promotes robust axonal regeneration after optic nerve injury [168]. In turn, PTEN opposes phosphatidylinositol 3-kinase (PI3K) by converting phosphatidyl inositol 3,4,5 trisphosphate (PIP_3_) to phosphatidyl inositol 4,5 bisphosphate (PIP_2_), a process that is known to curtail axon regeneration [169]. Depletion of lipin1, an enzyme that coordinates the synthesis of glycerolipids, enhances axon regeneration after optic nerve injury [170]. Lipin1 upregulation after injury results in the production of triglycerides, rather than phospholipid membrane lipids. Thus, redirecting triglyceride to phospholipid synthesis by deleting diglyceride acyltransferases promotes axon regeneration [170].

After CNS damage multiple chemical and physical barriers prevent axons to regrow and colonize the affected area. Many of these molecules and their receptors have been shown to be associated with lipid rafts, as an essential feature for their functionality. Lipid rafts can be specifically targeted with pharmacological reagents, and it has been demonstrated that lowering cholesterol levels, the principal component of lipid raft microdomains, enhances axonal regeneration in both the CNS and the PNS [148,149,150,151,152,153,154,171]. Thus, local destabilization of lipid rafts after CNS injury may be a promising therapy to promote axonal regeneration. Lipid rafts are essential for the proper operation of the CNS, and they are important for the stability of synapses, endo- and exocytosis and communication with the extracellular environment. Consequently, lipid rafts should be modified exclusively at the site of injury in order to preserve their optimal functions in the nervous system. Adopting biomaterials capable of gradually release lipid raft modifying drugs may be a promising strategy to efficiently promote functional axon regeneration and reinnervation after damage. 

## Figures and Tables

**Figure 1 ijms-22-05009-f001:**
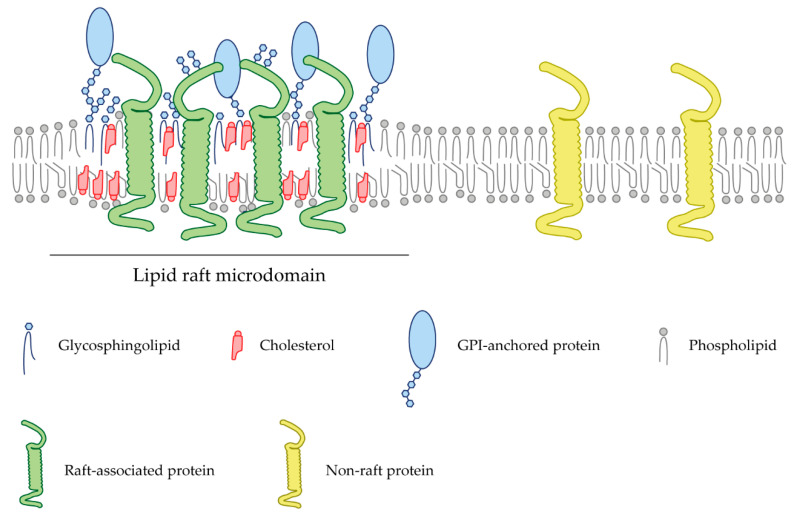
Schematic representation of a lipid bilayer of a cell membrane depicting lipid raft-associated proteins (green) and transmembrane proteins (yellow) excluded from these domains. Lipid rafts are membrane fractions enriched in cholesterol (red) and sphingolipids (blue), as well as harboring GPI-anchored proteins (light blue).

**Figure 2 ijms-22-05009-f002:**
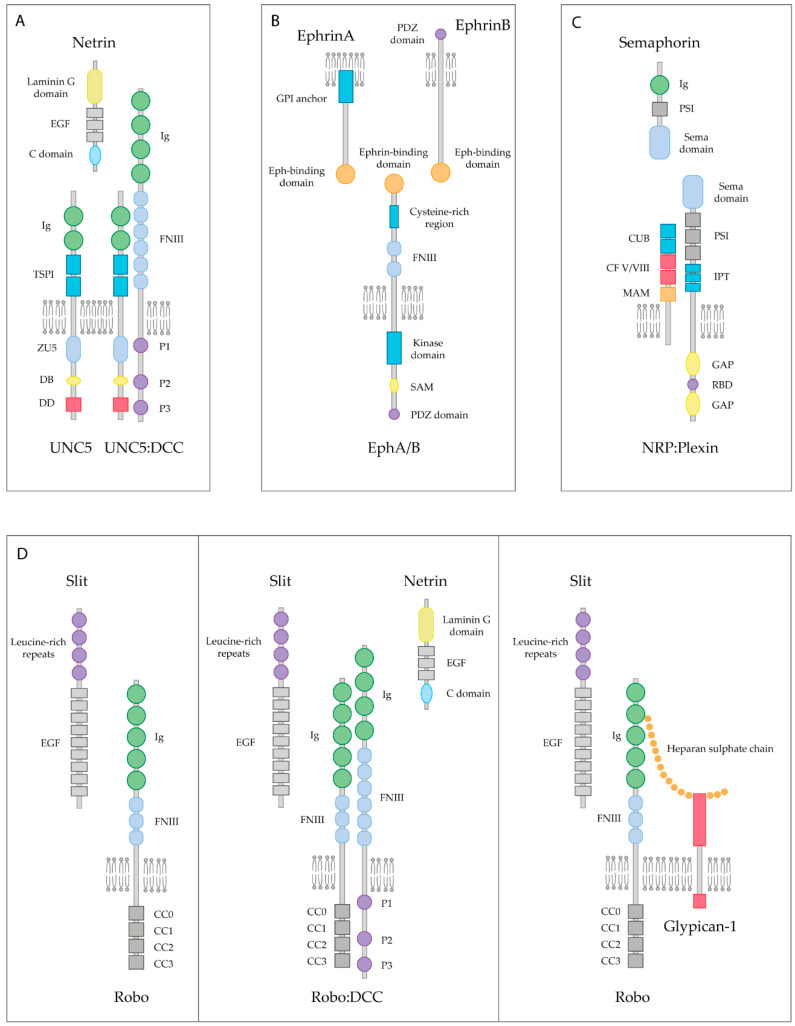
Lipid raft-dependent inhibitory axonal growth signaling of guidance cues and their receptors. (**A**) Axonal growth inhibition exerted by Netrin occurs in the presence of UNC5 receptors, either when bound to UNC5 alone or when heterodimerizing with the DCC receptor. Ig: Immunoglobulin domain (green); TSPI: Thrombospondin type I domain (blue); ZU5: Zona Occludens (light blue); DB: DCC-binding domain (yellow); DD: Death Domain (red); FNIII: Fibronectin type III domain (light blue); EGF: Epidermal Growth Factor repeats (gray). (**B**) Eph transmembrane receptors bind to both, GPI-anchored ephrinA or ephrinB, resulting in contact-mediated repulsion. PDZ: Postsynaptic Density 95-Discs large-Zonula occludens-1-protein (purple); SAM: Sterile Alpha Motif (yellow). (**C**) Semaphorin first interacts with the NRP receptors, which is the initial step for a receptor complex assembly. Transmembrane Semaphorins are also recognized by plexins, which are their binding receptor. PSI: Plexin-Sema-Integrin domain (gray); CUB: Complement-like domain (blue); CF V/VIII: V/VIII Clotting Factor domain (red); MAM: Meprin-like domain (orange); IPT: Immunoglobin-like fold shared by plexins and transcription factors (blue); GAP: GTPase Activating Protein domain (yellow); RBD: Rho-binding domain (purple). (**D**) Slit is able to bind the Robo receptor, which in turn inactivates intracellular machinery blocking the axonal extension. In the presence of both Slit and Netrin, the Robo and the DCC receptors interact, causing the inhibition of chemoattraction expected by the Netrin-DCC pair. Heparan sulfate chains are co-receptors for Robo and Slit, which stabilize the Slit-Robo complex, resulting in axonal repulsion.

**Table 1 ijms-22-05009-t001:** Methodologies used to study lipid-raft-associated proteins.

Method	Type	Description	References
Membrane fractionation	Biochemical	Solubilization of lipids and membrane proteins	[12,13]
Cholera Toxin B-subunit	Selective probe for membrane microdomains	Specifically binding to GM1	[14]
Laurdan	Environmentally sensitive fluorescence probe	Photoshifting between lipid-ordered and -disordered phases	[15,16]
Giant unilamellar vesicles	Artificial model membrane	Behavior of different lipid species	[17,18]
Fluorescence Recovery After Photobleaching (FRAP)	Biophysical	Study of lateral mobility of photolabeled proteins	[19,20]
Super-resolution microscopy	Biophysical	Visualization of molecules beyond the diffraction limit	[21]
Fluorescence Resonance Energy Transfer (FRET)	Biophysical	Visualization of dynamic signaling events at membrane microdomains	[22,23]

**Table 2 ijms-22-05009-t002:** Lipid raft destabilizing agents.

Agent	Type	Mechanism	References
Cyclodextrins	Cyclic oligosaccharides	Cell membrane cholesterol depletion	[135]
Statins	HMG-CoA inhibitor	Cholesterol biosynthesis inhibition	[136,137]
Cholesterol Oxidase	Enzymatic flavoprotein	Cholesterol oxidation	[138,139]
Filipins	Polyene macrolides	Cell membrane cholesterol sequestration	[140,141]
Apolipoprotein A-I Binding Protein	Protein encoded by the *APOA1BP* gene	Cholesterol efflux promotion	[142]
Triparanol	Δ^7^-Reductase inhibitor	Cholesterol biosynthesis inhibition	[136]
Glycophosphatidylinositol-specific Phospholipase C	Enzymatic phospholipase	Cleaving of GPI-anchored surface proteins	[143,144]
Overexpression of Cyp46A1	Cholesterol-catabolic enzyme	Cholesterol conversion to (24S)-24-hydroxycholesterol	[145]
Fumonisin B1	Sphingolipid synthesis inhibitor	Synthesis inhibition of dihydroceramide	[146,147]

**Table 3 ijms-22-05009-t003:** Studies of cholesterol-lowering drugs in axonal regeneration models.

Compound	Model	Effect	References
MβCD and ChOx	Temporal explants (in vitro)	Axonal growth on inhibitory RGMa substrate	[148]
MβCD	Axotomized retinal ganglion cells (in vivo)	Neogenin-dependent apoptosis blocking	[148]
MβCD	SCI (in vivo)	Motor functional recovery	[148]
Lovastatin	Optic nerve injury (in vivo)	Regeneration and axonal sprouting	[150]
Prominin-1 overexpression	Cultured DRG neurons (in vitro)	Enhanced axonal regeneration	[151]
Statins	Spinal motor neurons (in vitro)	Axonal growth on inhibitory substrates	[152]
Cerivastatin	Optic nerve injury (in vivo)	Regeneration of retinal ganglion cell axons	[152]
MβCD, ChOx and Nystatin	CNS and PNS developing neurons (in vitro)	Increased growth cone area, density of filopodium-like structures and the number of neurites branching points	[153]
Nystatin	Axotomized hippocampal neurons in a microfluidic chamber (in vitro)	Increased axonal regeneration	[153]
Nystatin	Axotomized entorhinal-hippocampal projection in organotypic culture (ex vivo)	Increased axonal regeneration	[153]
MβCD	Sciatic nerve injury (in vivo)	Axonal regeneration and functional recovery	[153]
2-hydroxypropyl-β-cyclodextrin	SCI (in vivo)	Increased axon regeneration	[154]

## Data Availability

Not applicable.

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
