# Peer review of "One Raft to Guide Them All, and in Axon Regeneration Inhibit Them"

_ijms, 2021, doi:10.3390/ijms22095009_

Round 1

Reviewer 1 Report

No specific comments for the authors.

Author Response

No comments from Reviewer 1. Therefore, no further response is required.

Reviewer 2 Report

The review is about lipids rafts in axon regeneration. The majority of the literature review covers extrinsic factors limiting repair and how their receptors are dependent on lipids. The review ends with a section about lowering cholesterol promoting axon regeneration.

The strong points of this review are:
+ Section number 5 with examples of lowering cholesterol to promote axon regeneration
+ All three tables are excellent (!) and will definitely be used by other scientist in presentations and will also generate citations for this article.
+ A large range of extracellular factors is discussed and summarized in a figure.

I do recommend revisions for the review, because some literature seems to be lacking in the current version. I hope the suggestions will improve the manuscript further. It is good to have a review with a lipid raft perspective on axon regeneration.

  1. Regarding the title “One raft to guide them all, and in regeneration inhibit them”.
    A) Are you suggesting that lipid rafts are all the same (see also comment 2 below)?
    B) Are you suggesting that lipid rafts on neurons are detrimental for its intrinsic regeneration capacity and therefore indeed need to be inhibited? Lines 504-507 in the conclusion section state that it is unclear whether to inhibit or stimulate of lipid drafts for axon regeneration. Please clarify. The authors can decide to keep their title though, as it is a matter of personal preferences.

  2. Below questions remain unanswered in this review. Perhaps you could consider your responses for the review?
    A) Is it known whether the composition of lipid rafts differ between PNS and CNS neurons (e.g. regenerating and non-regenerating neuron type)?
    B) Does the lipid raft composition differ between immature and mature CNS neurons (see line 487)?
    C) Does the lipid draft composition differ between axons and dendrites as neurons are highly polarized with selective transport changes for receptors and most likely raft-associated proteins?
    D) Follow up on above questions: How would you target specific components of lipid rafts on adult CNS neurons to make them more corresponding to the immature / regenerating phenotype (lines 504 – 507)? Or are rafts indeed all the same as implied in the review title. I want to confirm here that the studies on lowering cholesterol are well described in the review and table 2 with lipid raft destabilizing agents is also very clear.

  3. In sections 2 till 4, I would recommend to, where possible, shorten the text about receptors and its ligands. The difference in regeneration potential between PNS and CNS should also be described in the introduction only, rather than mentioned at individual extrinsic factors (for example: Lines 152 – 157). This will increase the focus on receptors signaling in lipid rafts and further improve the merits of this review article.

  4. Sections 2 till 4 describe extrinsic factors regulating lipid rafts. Intrinsic factors regulating lipids and axon regeneration are not mentioned.
    A) Do you consider triglyceride (TG) and diglyceride acyltransferases (DGATs) part of lipid rafts? If yes, the paper about neuronal enzyme Lipin1, DGAT, and phospholipids in axon regeneration in retinal ganglion cells should be discussed. Liu laboratory.
    B) Figure 1 highlights phospholipids as part of lipid drafts. Are phosphatidylinositol 4,5-bisphosphate (PIP2) and phosphatidylinositol 3,4,5-trisphosphate (PIP3) considered part of lipid rafts? If this is the case, then literature on PTEN knockout and PI3K overexpression in the context of axon regeneration should be included in this review. It also has been reported that a developmental decline in phosphatidylinositol 3,4,5-trisphosphate contributes to axon regeneration failure in CNS neurons. He and Fawcett laboratories.

  5. Nogo (Lines 92; and 97 till 113) should not be described in section Section 2.1 Chondroitin Sulfate Proteoglycans. It should instead be part of the myelin-associated inhibitors.

  6. Please provide example of CSPGs and describe their role in axon repulsion in Section 2.1 Chondroitin Sulfate Proteoglycans. For instance, there is no literature on Aggrecan in the current version of this review. I would also recommend naming chondroitinase ABC enzyme (ChABC) in this section. You may also be interested to mention that glycan sulphation patterns (both CS and HS) influence autophagy, e.g. lipid metabolism, via PTPRσ receptors at the axon and regeneration. Kadomatsu laboratory.

  7. Please provide examples of HSPGs in Section 2.2 Heparan sulfate proteoglycans (HSPGs). For instance, there is no literature about Syndecan in axon regeneration mentioned.

  8. Section 2.3 Tenascins and section 3.1. Myelin-associated glycoprotein. Please mention that integrins are also receptors for myelin-associated glycoprotein and tenascin-C.

  9. Lines 128 – 130: Perineuronal nets are only found on a small proportion of CNS neurons. The current sentence implies that all CNS neurons have PNNs.

  10. Citations are absent but are needed in lines 212 till 217. Claims without reference include: “Nogo also interacts with integrins”; “Integrins are involved mainly in developmental processes”; “Integrins are highly down regulated in mature CNS”; “Integrins have been studied as pivotal players during axon regeneration in the CNS”; “PNS neuronal types that express relatively high levels of integrins”. Importantly, do you consider integrins part of lipid rafts or is it a non-raft protein? If integrins are considered a raft-protein then please state this. If a non-raft protein, then lines 212 till 224 could be shortened (see also comment 2)

  11. Consider removing the non-neuronal HEK 293T cells from table 3 about cholesterol lowering drugs in axon regeneration.

Author Response

Please see the attachment for Reviewer responses.

Round 2

Reviewer 2 Report

Thank you for your time and efforts on the revisions. The author responses and the changes made to the manuscript are clear. 

I recommend to accept in present form.